# Enhancing the Proton Exchange Membrane in Tubular Air-Cathode Microbial Fuel Cells through a Hydrophobic Polymer Coating on a Hydrogel

**DOI:** 10.3390/ma17061286

**Published:** 2024-03-11

**Authors:** Junlin Huang, Chih-Hung Wu, Fuying Li, Xiang Wang, Sheng-Chung Chen

**Affiliations:** 1College of Chemical Engineering, Fuzhou University, Fuzhou 350116, China; jlfz2024@163.com; 2School of Resources and Chemical Engineering, Sanming University, Sanming 365004, Chinabenbear.xe@gmail.com (S.-C.C.); 3Fujian Provincial Key Laboratory of Resources and Environment Monitoring & Sustainable Management and Utilization, Sanming University, Sanming 365004, China; wangxiang961025@163.com; 4State Key Laboratory of Photocatalysis on Energy and Environment, Fuzhou University, Fuzhou 350108, China; 5Department of Engineering Technology Management, International College, Krirk University, Bangkok 10220, Thailand

**Keywords:** hydrophobicity, coating, hydrogel, microbial fuel cells, proton exchange membrane

## Abstract

The usage time of air-cathode microbial fuel cells (MFCs) is significantly influenced by the moisture content within the proton exchange membrane (PEM). Therefore, enhancing the water retention capability of the PEM by applying a hydrophobic polymer coating to its surface has extended the PEM’s usage time by three times and increased MFCs’ operational duration by 66%. Moreover, the hydrophobic nature of the polymer coating reduces contamination on the PEM and prevents anode liquid from permeating into the air cathode. Towards the end of MFC operation, the internal resistance of the MFC is reduced by 45%. The polymer coating effectively maintained the oxygen reduction reaction activity in the cathode. The polymer coating’s ability to restrict oxygen transmembrane diffusion is demonstrated by experimental data showing a significant decrease in oxygen diffusion coefficient due to its presence. The degradation efficiency of the chemical oxygen demand from 16% to 35% increased by a factor of one.

## 1. Introduction

Microbial fuel cells (MFCs), as a novel technology capable of simultaneous wastewater treatment and energy recovery, have garnered increasing attention [1,2,3]. This is because they harness microbial degradation of organic compounds, converting chemical energy into electrical energy [4,5,6]. Inside the MFC, organic matter is degraded by microorganisms, generating electrons at the anode. These electrons are then transferred through a proton exchange membrane (PEM) in the form of protons to reach the cathode. Simultaneously, an electric current is produced in the external circuit between the anode and cathode, forming a complete electrical circuit with the internal components of the MFC.

The PEM is one of the core components of the MFC and effectively allows selective permeation of protons [7,8]. In air-cathode microbial fuel cells, the PEM prevents prolonged direct contact between the air cathode and the anode liquid, thereby reducing the formation of biofouling on the air cathode and subsequently diminishing the kinetics of the oxygen reduction reaction (ORR) on the air cathode, consequently affecting the MFC’s performance [9]. The utilization of oxygen by bacteria as an electron acceptor for oxidation is a key factor contributing to the diminished coulombic efficiency (CE) observed in air-cathode microbial fuel cells. This phenomenon, as reported by [10], leads to the loss of substrate. The PEM can prevent oxygen diffusion from the air cathode to the anode, thus preserving the anaerobic environment in the anode chamber. Studies have reported the use of various separation membranes to prevent oxygen diffusion from the cathode to the anode, including ultrafiltration membranes [11], permeable membranes [12], cathode ion exchange membranes [13], and modifications to the air cathode to limit oxygen diffusion [14,15].

Protons primarily transport within the PEM using water molecules as carriers [16,17]. Therefore, the efficiency of proton transport within the PEM strongly relies on its water content. Polyvinyl alcohol (PVA) is a water-soluble polymer renowned for its exceptional mechanical properties, water absorption capacity, film-forming prowess, non-toxic nature, and cost effectiveness, as evidenced by various studies [18,19,20]. It can potentially serve as a PEM in MFC systems [21,22,23]. When the PVA hydrogel (PVA-H) is used as a PEM due to its hydrophilic surface, it is easy to absorb the sludge in the anodic solution, resulting in the scaling phenomenon of the PEM during the operation of the MFC [24], and the sludge in the anodic solution will reach the air cathode with the transmission of protons. Biological fouling on the air cathode [9] reduces the ORR activity of the air cathode, which will greatly reduce the electric generation performance of the MFC. In addition, when PVA-H are exposed to air, the water inside the hydrogels is easy to evaporate, resulting in a decrease in proton transport efficiency [25], which results in a decrease in the voltage of the MFC. Therefore, improving the water retention capacity of PVA-H as proton membranes of the MFC is a key challenge. One possible way to increase water content in proton-conducting polymers is their modifying with inorganic ion exchangers. However, their effect is ambiguous; the modifier can both increase and reduce water uptake. This behavior is caused by morphology of embedded inorganic particles.

Constructing a coating on hydrogels is a method to prevent dehydration. It can avoid embedding inorganic particles inside the ion-exchange polymer. Similar to mammalian skin, it prevents rapid evaporation of moisture from within cells. Taking inspiration from this, Mredha et al. [26] initially adsorbed the initiator azobisisobutyronitrile (AIBN) onto the hydrogel surface. Then, they triggered the polymerization of macromolecular monomers at 120 °C, forming a polymer coating on the hydrogel surface. Subsequently, the hydrogel was immersed in silicone oil at 120 °C to create an oil layer and finally quenched in silicone oil at 20 °C. After modification, the weight decreased by only 4.1% within 7 days at 25 °C. Similarly, Zhu et al. [27] treated the surface of the polyacrylamide hydrogel (PAAM-H) with plasma to generate active hydroxyl groups. Using consecutive impregnation methods, they constructed a dual hydrophobic coating of stearic acid (STA) and silicone oil (SO) on the PAAM-H surface. Within 5 days, the dual hydrophobic-coated hydrogel retained over 70% of its moisture content. Gao et al. [28] added polystyrene sulfonate sodium (PSS) into PAAM-H precursor solution, and then soaked the prepared hydrogel in cationic-modified gelatin (CG) solution; CG and PSS will be firmly bonded to the surface of the hydrogel through strong electrostatic force interaction. Subsequently, the hydrogel underwent rapid quenching by immersion in glycerin at −20 °C. This process induced the “sol-gel transition” of gelatin, resulting in the formation of a water coating composed of gelatin and glycerin on the hydrogel’s surface. Remarkably, the weight retention rate reached 72.5% after exposure to conditions of 25 °C and 40% relative humidity for a period of 5 days.

The surface of PVA-H itself is hydrophilic, resulting in weak bonding with the hydrophobic coating. Silane coupling agents possess dual-affinity characteristics [29,30,31]. In this study, 3-aminopropyltrimethoxysilane (APTMS) was used as a connecting bridge, while behenic acid (BA) served as the hydrophobic material to construct a hydrophobic polymer coating on the surface of PVA-H. This coating was intended to serve as the PEM in tubular air-cathode microbial fuel cells (TAC-MFCs). The aim of this study was to investigate the effects of the polymer coating on the PEM and the usage time of the MFC, the level of membrane contamination, ORR activity on the air cathode, and oxygen diffusion degree, and the degradation efficiency of chemical oxygen demand.

## 2. Materials and Methods

### 2.1. Preparation of Polymer Coating

The materials used in this experiment, 3-aminopropyltrimethoxysilane (APTMS), behenic acid (BA), all analytical grade, were purchased from Aladdin (Shanghai, China). n-Hexane, of analytical grade, was purchased from Xilong Chemical Co., Ltd. (Shantou, China). PVA 2699 was purchased from Anhui Wanwei Group Co., Ltd. (Chaohu, China). Additionally, the 30 W UV-O_3_ lamp with a wavelength of 254 nm and 185 nm was acquired from Guangzhou Xuelai Te Company (Guangzhou, China).

Modification process: This experiment involves first performing surface treatment on the PVA hydrogel and then preparing a polymer-coated hydrogel through a continuous immersion method. The specific steps are as follows: (1) Pour 10% PVA water solution into a cylindrical mold with a lid and a diameter of 1.5 cm. Freeze at −20 °C for 20 h, thaw for 1 h, and repeat this cycle five times. Before the PVA hydrogel thaws, use an iron rod to push the cylindrical hydrogel out of the mold, and cut the thawed cylindrical PVA hydrogel into 1 cm lengths; (2) Prepare a 7% APTMS water solution, ensuring complete hydrolysis. Simultaneously, prepare a 0.3 mol/L solution of BA/n-Hexane; (3) Wipe off the surface moisture of the PVA hydrogel and then immediately immerse it in the APTMS water solution for a period after irradiation under a UV-O_3_ lamp at wavelengths of 254 nm and 185 nm for 15 min; (4) Take out the PVA hydrogel soaked in the APTMS water solution; immediately immerse it in the BA/n-Hexane solution after removing excess water from the hydrogel surface; (5) Retrieve the hydrogel from the BA/n-Hexane solution; allow it to stand in the air for some time, enabling excess solvent on the hydrogel surface to evaporate. This yields an AB-PVA-H (Figure 1a).

Modification principle: Due to the fact that the side chain of the PVA hydrogel itself contains abundant hydroxyl groups, which can form hydrogen bonds with silicon hydroxyl groups generated after the hydrolysis of APTMS, the number of active hydroxyl groups can be increased by irradiating the surface of the PVA hydrogel with UV-O_3_ lamp. The amino groups at the other end of APTMS and carboxyl groups in BA are combined by covalent bonding, which grafts hydrophobic long-chain fatty acids onto the hydrophilic PVA hydrogel surface (Figure 1b), forming a dense hydrophobic polymer coating on the surface of the PVA hydrogel.

### 2.2. Characterization

The contact angle of the hydrogel surface was measured at room temperature using a contact angle measurement instrument (JC2000D1, Shanghai Zhongchen Digital Technology Equipment Co., Ltd., Shanghai, China) via the sessile drop method. The surface and cross-sectional structures of the hydrogel coating were observed using a scanning electron microscope (SEM; JSM-IT500LA, Tokyo, Japan) at an acceleration voltage of 15.0 kV. Fourier transform infrared spectroscopy (FTIR, IRAffinity-1S, Tokyo, Japan) was employed to analyze the functional group structure of the surface coating after PVA-H modification. The coating was ground into a dry powder and placed on an ATR platform for scanning, with settings of 20 scans, a scanning wavelength range of 400–4000 cm^−1^, and a measurement mode set to transmittance. Dissolved oxygen concentration was measured by the dissolved oxygen meter (DO Meter, AR8604, SMART, Dongguan, China).

### 2.3. MFC Setup and Operation

The structure of the cathode anode and proton exchange membrane is shown in Figure 2a. The MFC consists of a PVC cylindrical shell (inner diameter of 16 mm and length of 20 cm) with perforations on the lower half of the shell. Inside the tube, the bottom portion serves as the anode (2 g stainless wool), followed by the PEM (a PVA-H with a diameter of 16 mm and a length of 8 cm) extending to the top of the tube. The graphite rod (inner diameter of 5 mm, length of 5 cm) is embedded in the proton membrane by approximately 1 cm. Approximately 300 mL of anaerobic bacterial sludge culture was mixed with artificial simulated wastewater at a ratio of 2:1 and introduced into the anode chamber (a 330 mL plastic bottle). A hole was drilled in the bottle cap to insert the TAC-MFC anode end into the bottle, while the cathode end was exposed outside the bottle. The height of the anode liquid submerged the perforated area of the anode section, as depicted in Figure 2b.

TAC-MFC has an external test resistance of 1000 Ω, which tests the voltage output of the battery through the data collector under the load of 1000 Ω. The entire operation process is divided into three stages: early stage (Stage I), middle stage (Stage II), and late stage (Stage III), observed for the MFC’s power generation performance and the microbial degradation of organic matter. These stages are denoted as Stage I (voltage rising from 550 mV to 650 mV), Stage II (voltage decreasing from 400 mV to 300 mV), and Stage III (voltage decreasing from 200 mV to 100 mV). During Stage I, after introducing the anode liquid, the initial chemical oxygen demand (COD) was measured. Then, the voltage gradually increased from the initial value to reach a maximum of 600 mV. At this point, the polarization curve, power density, and COD value of the TAC-MFC were measured. The polarization curve and power density curve were obtained by varying the external resistance. The 1000 Ω resistance were replaced with a variable resistance box, and the maximum power density can be determined by adjusting the resistance value of the resistance box (99,999~100 Ω); at this time the internal resistance of the MFC and the number on the resistance box are equal. In Stage II, the voltage began to decrease, and measurements of COD, polarization curve, and power density of the MFC were taken. By Stage III, the voltage had reduced to 100 mV, and measurements of COD, polarization curve, and power density were conducted. Photographs of the PEM and the cathode were taken at each stage to observe the degree of contamination. Electrochemical techniques including cyclic voltammetry (CV) were employed to assess ORR activity, and electrochemical impedance spectroscopy (EIS) was used to evaluate changes in the MFC’s internal resistance at each stage.

To evaluate the impact of hydrophobic polymer-coated hydrogels in mitigating oxygen diffusion, a device simulating oxygen permeation in the MFC operating environment (Figure 3) was used. Dissolved oxygen (DO) concentration curves in the anode chamber liquid were plotted, and the oxygen transfer coefficient of the PEM was calculated to assess the effect of hydrophobic polymer-coated hydrogels in reducing oxygen diffusion.

### 2.4. Measurement and Calculation Method

The modified and unmodified PVA-H were exposed to the environment with a relative humidity (RH) of 30% and a temperature (T) of 25 °C, and the hydrogel’s mass was measured daily, along with photographs taken of its appearance, following Equation (1) [28], to assess the hydrogel’s preservation before and after modification:(1)$${WR}_{i}=W_{i}/W_{0}$$
where *WR_i_* represents the water retention rate. *W_i_* (g) and *W*_0_ (g) denote the weight of the hydrogel at time *i* (day) and time 0, respectively.

The output voltage of the TAC-MFC was recorded at intervals of 5 s using a data acquisition system (PS-DAQ, Smacq, Beijing, China). The COD of the anode liquid was determined through the fast digestion–spectrophotometric method: multifunctional fast digester (GL-25K, China-GLKI, Beijing, China), multi-parameter water quality analyzer (GL200, China-GLKI, Beijing, China). COD degradation efficiency (ηCOD) was calculated using Equation (2):(2)$$\eta_{COD}=\frac{C_{0}-C_{t}}{C_{0}}\times 100\%$$
where C_0_ (mg L^−1^) represents the initial COD concentration of the anode liquid, and *C_t_* (mg L^−1^) represents the COD concentration of the anode liquid at time *t* (s).

The data associated with the ORR at the cathode and the charge transfer resistance related to the proton membrane’s contamination were gathered using an electrochemical workstation (CHI604e, Chenhua, Shanghai, China), using cyclic voltammetry (CV) and electrochemical impedance spectroscopy (EIS) curves for determination. In these tests, the cathode acted as the working electrode, the anode as the counter electrode, and a saturated calomel electrode was employed as the reference electrode (RE) relative to the standard hydrogen electrode. CV curves were recorded from 0.6 V to −0.6 V at a scan rate of 50 mV/s. EIS tests were conducted on the entire MFC system with a scan frequency range from 0.01 to 1M Hz.

The oxygen permeation device utilized a 1 L serum bottle to simulate the anode chamber of MFC. A hole was made in the rubber stopper of the serum bottle to insert a DO probe and a PVC tube. Subsequently, the crevices around the rubber stopper were wrapped with sealing film (Parafilm, Wisconsin, Neenah, USA) to prevent air from seeping into the anode chamber. The oxygen diffusion coefficient of the membrane was calculated using Equation (3) [10] to assess the effect of the hydrophobic polymer-coated hydrogel on reducing the extent of oxygen diffusion:(3)$$k=-\frac{V}{At}ln\left[\frac{C_{s}-C_{t}}{C_{s}}\right]$$

Here, *k* (cm s^−1^) is the oxygen diffusion coefficient and V (0.33 L) represents the volume of the anode chamber. A is the working area of the cathode (1.77 cm^2^), *C_t_* (mg L^−1^) is the oxygen concentration in the body solution at time *t* (s), and C_s_ (310 mg L^−1^) is the concentration on the air side of the cathode (assumed to be the saturation concentration of oxygen in water, with a value of 7.8 mg L^−1^). Before we start the experiment, given the provided information about the DO probe fixation in the center of the anode liquid, the placement of a magnet at the base for uniform microbial distribution, the use of a magnetic stirrer for stirring, and the deoxygenation of the system with gaseous nitrogen until the DO concentration drops below 0.1 mg L^−1^, it indicates a procedure ensuring a low-oxygen environment for the microbial fuel cell experiment (Figure 3).

## 3. Results and Discussion

### 3.1. Structure and Morphology Characterization

This process generates a dense hydrophobic polymer coating on the surface of the PVA-H. The FT-IR (Figure 4) spectrum of the APTMS-BA coating exhibits the disappearance of the peak at 1695 cm^−1^ corresponding to the COOH stretching vibration of BA, and the appearance of peaks corresponding to amide I (υ (C=O)) and amide II (δ (N-H)) at 1700 cm^−1^ and 1540 cm^−1^, respectively. This indicates the formation of an amide bond between BA and APTMS, confirming the successful grafting of BA onto APTMS. These experimental findings are consistent with Si et al. [32].

The surface and cross-section of the modified hydrogel were characterized using scanning electron microscopy (SEM), revealing a relatively dense structure on the coating surface (Figure 5a). This density is advantageous in preventing the evaporation of moisture within the hydrogel. The cross-sectional SEM image illustrates a tight bond between the coating and hydrogel surfaces (Figure 5b), indicating a strong adhesion between the polymer coating and the hydrogel without significant gaps at the interface. This robust adhesion implies that the surface coating is firmly attached to the hydrogel and unlikely to detach easily.

The cross-sectional analysis also reveals that the contact angle of the PVA-H surface underneath the coating is 14.84°, whereas the coating surface exhibits a contact angle of 154.34°. This shift indicates a transformation of the modified PVA-H surface from hydrophilic to hydrophobic. The high hydrophobicity can be attributed to the reaction between BA and APTMS, resulting in a block-like structured long chain with low surface tension and binary microstructure, effectively reducing the contact area between the coating surface and liquids [28,32].

### 3.2. Water Retention Test for Hydrogels

After the modification of the hydrogel, adjusting the modification time becomes a crucial factor in water retention [28]. The change of the weight over time for the hydrophobic polymer-coated hydrogel can be defined as A_x_B_y_-PVA-H. (where A_x_ represents the modification time of PVA-H in APTMS/H_2_O solution, and by represents the modification time in BA/n-Hexane solution, fundamental unit for minute.) Upon comparing multiple sets of data (Figure 6a,b), it was observed that the A_20_B_40_-PVA-H combination exhibited the strongest water retention capability.

Calculations based on Equation (1) revealed a water retention rate of 76% over five days for this specific combination. The contact angle for this hydrogel was measured at 154.34°. In Figure 6b, the A_20_B_60_ group exhibited inferior water retention than the A_20_B_40_ group, likely due to the extended modification time causing the coating to become excessively thick, making it prone to detachment from the hydrogel surface, thereby affecting its water retention capability. Therefore, the A_20_B_40_-PVA-H (AB-PVA-H) was selected as the PEM for TAC-MFC performance testing.

It is evident that the polymer coating after modification effectively delayed the internal moisture loss of the hydrogel (Figure 6c). The PVA-H lost all its moisture in just four days when exposed to room temperature air, while the AB-PVA-H took 16 days to completely lose its moisture. The modified polymer coating extended the water retention time of the hydrogel to three times longer than the original. Additionally, combining observations from Figure 6d, it was noted that by the fifth day, the volume of the original hydrogel had shrunk to approximately 1/4 of its initial size when completely dried. However, the volume of AB-coated H is almost the same as the initial state, which indicates that the hydrophobic polymer coating can improve the water retention performance of the hydrogel.

### 3.3. Voltage Output of Tubular Air-Cathode Microbial Fuel Cells

The comprehensive voltage output (Figure 7) of both PVA-H and AB-PVA-H, serving as the proton exchange membrane (PEM) in TAC-MFC (denoted as PVA-TAC-MFC and AB-TAC-MFC), respectively, is depicted across various stages. In Stage I, the voltage of the MFC starts at approximately 100 mV. This initial difference in potential arises due to the cathode using a graphite rod and the anode utilizing steel wool, resulting in a potential difference. Upon introducing the anolyte, the microbial action begins to generate electricity in the MFC. After approximately 24 h, both reach a maximum voltage of approximately 600 mV and fluctuate around this value before stabilizing.

During Stage II, the PVA-TAC-MFC exhibits a rapid decline in voltage due to the quick dehydration of the proton-exchange membrane, leading to a rapid decrease in proton transfer rate [33]. Conversely, in the AB-TAC-MFC, the presence of the polymer coating delays the rate of membrane dehydration, resulting in a slower decline in voltage compared with the PVA-TAC-MFC. Moving to Stage III, the voltage of the PVA-TAC-MFC fluctuates at approximately 200 mV, whereas the voltage of the AB-TAC-MFC continues a gradual decline as seen in the previous trend. This disparity is due to the PEM of the PVA-TAC-MFC becoming completely dry at the cathode end, reducing contact area and resulting in unstable voltage output. Meanwhile, the presence of the polymer coating in the AB-TAC-MFC prevents complete dehydration of the hydrogel, ensuring good contact between the PEM and cathode, leading to a relatively stable voltage trend.

Eventually, both voltages reach approximately 100 mV, similar to the initial voltage; at this point, both proton membrane cathodes are completely dry, signifying hindered proton transfer between the cathode and anode [34]. Over the entire process, the PVA-TAC-MFC runs for 265 h from start to finish, whereas the AB-TAC-MFC operates for approximately 440 h, extending the MFC’s usage time by approximately 66% (Figure 7).

### 3.4. Impact of Hydrophobic Polymer Coating on the Kinetics of Cathodic Oxygen Reduction Reaction

Due to the hydrophobic nature of the polymer coating, it prevents the infiltration of anolyte into the air cathode [35]. In Figure 8a, Stage I, there is minimal contamination on the PEM and air cathode for both PVA-TAC-MFC and AB-TAC-MFC. According to the CV curve (Figure 8b), PVA-TAC-MFC cathodic reduction peak potential is approximately −0.28 V, reaching a peak current of approximately −0.53 mA. As seen in the power density plot in Figure 8d, PVA-TAC-MFC achieves a maximum power density of 1.96 mW/m^2^, whereas AB-TAC-MFC shows a cathodic reduction peak potential of approximately −0.27 V (Figure 8c), a peak current of −0.4 mA, and a slightly lower maximum power density of 1.81 mW/m^2^ (Figure 8e) compared with PVA-TAC-MFC. The internal resistance of PVA-TAC-MFC and AB-TAC-MFC can be calculated as 3438.7 Ω and 5512.6 Ω, respectively, after fitting the polarization curve. This is due to the intrinsic resistance of the polymer coating, mildly compromising the initial ORR activity of the cathode.

In Stage II, a substantial amount of anolyte sludge adheres to the surface of the PEM (Figure 8a, Stage II) in PVA-TAC-MFC, infiltrating the cathode and forming biofouling. With the peak current decreasing to −0.23 mA (Figure 8b), this indicates hindrance caused by biofouling to the cathodic ORR reaction, akin to findings by Li et al. [9]. The power density drops to 0.82 mW/m^2^ (Figure 8d). Conversely, in AB-TAC-MFC (Figure 8a, Stage II), the strong hydrophobicity prevents anolyte sludge from infiltrating the cathode, preventing biofouling formation. With a peak current of −0.33 mA (Figure 8c) and a higher power density of 1.25 mW/m^2^ (Figure 8e), by fitting the polarization curve, the internal resistance of PVA-TAC-MFC and AB-TAC-MFC at this stage is 7853.2 Ω and 6615.6 Ω, respectively. This is because the proton membrane of PVA-TAC-MFC has been polluted at this time, and the proton membrane is dry to a large extent, which increases the resistance of proton transfer and leads to the increase in the internal resistance of the battery. ORR activity was lower than that of AB-TAC-MFC.

By Stage III, the PEM of PVA-TAC-MFC is heavily contaminated, and significant biofouling is present on the cathode surface (Figure 8a-Stage III), resulting in a loss of oxidation peaks and slight reduction peaks. The peak current reduces to nearly 0.2 mA (Figure 8b), and the power density declines to 0.11 mW/m^2^ (Figure 8d). AB-TAC-MFC shows increased membrane contamination at this stage (Figure 8a, Stage III), but the cathode remains free from sludge. The peak current is approximately 0.25 mA (Figure 8c), with a power density of 0.43 mW/m^2^ (Figure 8e). At this time, the internal resistance of AB-TAC-MFC is 17,303 Ω, which is 45% lower than that of PVA-TAC-MFC (31,402 Ω), indicating that the polymer coating can reduce the degree of contamination of the proton membrane, thereby reducing the internal resistance of the MFC [24]. 

During the MFC operation, changes in internal resistance can be characterized using EIS. According to Figure 8f, it is observed that in the initial stages of the MFC’s operation, the arc radius of PVA-TAC-MFC is smaller than that of AB-TAC-MFC, with a charge transfer resistance (R_ct_) of approximately 1.5 kΩ, while the R_ct_ of AB-TAC-MFC is approximately 2.5 kΩ. The presence of the surface polymer coating in AB-TAC-MFC moderately increases R_ct_, enlarging the MFC’s impedance. However, towards the end of the MFC operation, as the PEM dries out, the impedance of both cells increases. For PVA-TAC-MFC, with its membrane being contaminated with biofouling on the cathode, R_ct_ rises to approximately 160 kΩ, while for AB-TAC-MFC, R_ct_ is approximately 30 kΩ, which is approximately 1/5 of PVA-TAC-MFC’s. This demonstrates that the hydrophobic groups in the polymer coating reduce the membrane contamination, leading to decreased charge transfer resistance in the MFC.

### 3.5. Transmembrane Diffusion of Oxygen

According to Figure 9, the extent of transmembrane diffusion of oxygen before and after the proton membrane modification can be observed. It is evident that the oxygen diffusion curve of AB-TAC-MFC is gentler compared with that of PVA-TAC-MFC. Calculating from Equation (3), the oxygen diffusion coefficient for PVA-TAC-MFC is determined to be 2.402 ± 0.005 cm s^−1^, whereas for AB-TAC-MFC, it is relatively smaller at 1.859 ± 0.002 cm s^−1^ (Table 1). This is in line with the findings of [10] and others, where the data for both cases fall between the oxygen diffusion coefficients of the blank and control groups. This suggests that the polymer coating can restrict oxygen diffusion from the cathode to the anode chamber, thus maintaining a better anaerobic environment within the anode chamber.

### 3.6. Degradation Efficiency of Chemical Oxygen Demand

Table 2 illustrates the degradation efficiency of COD concentration throughout the entire operational phase of the MFC. Calculated using Equation (2), ηCOD of the PVA-TAC-MFC is only 16%, while that of the AB-TAC-MFC reaches 35%, increased by a factor of one.

## 4. Conclusions

This study, through modifying the PEM of TAC-MFC to create a hydrophobic polymer coating, summarizes the effects of the hydrophobic polymer coating: it extends the usage time of both the PEM and MFC, reduces the pollution levels on the PEM and air cathode, lowers the internal resistance of the MFC, and restricts oxygen diffusion from the air cathode to the anode chamber, thereby maintaining an anaerobic environment and improving the degradation efficiency of COD. According to experimental data,
The polymer coating extends the PEM usage time by three times, increasing MFC runtime from the original 265 h to 440 h, an extension of approximately 66%;Hydrophobicity in the polymer coating reduces PEM contamination levels and reduces the internal resistance of the MFC. At the end of the MFC operation, the internal resistance of the MFC is reduced by 45%. The polymer coating prevents the anode liquid from penetrating into the cathode along the PEM surface, reduces the formation of biological fouling on the cathode, and effectively maintains the ORR activity of the cathode;The polymer coating reduces the degree of oxygen transmembrane diffusion to the anode chamber. The oxygen permeability coefficient is significantly lower compared with the PEM without the polymer coating. The degradation efficiency of COD from 16% to 35% increased by a factor of one.

## Figures and Tables

**Figure 1 materials-17-01286-f001:**
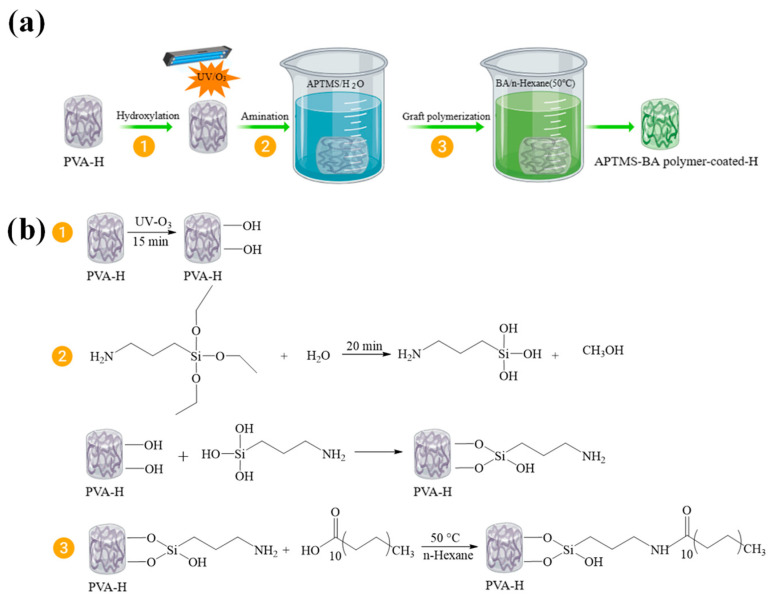
(**a**) Schematic diagram for PVA-H modification; (**b**) flowchart outlining the three-step process for the modification of PVA-H.

**Figure 2 materials-17-01286-f002:**
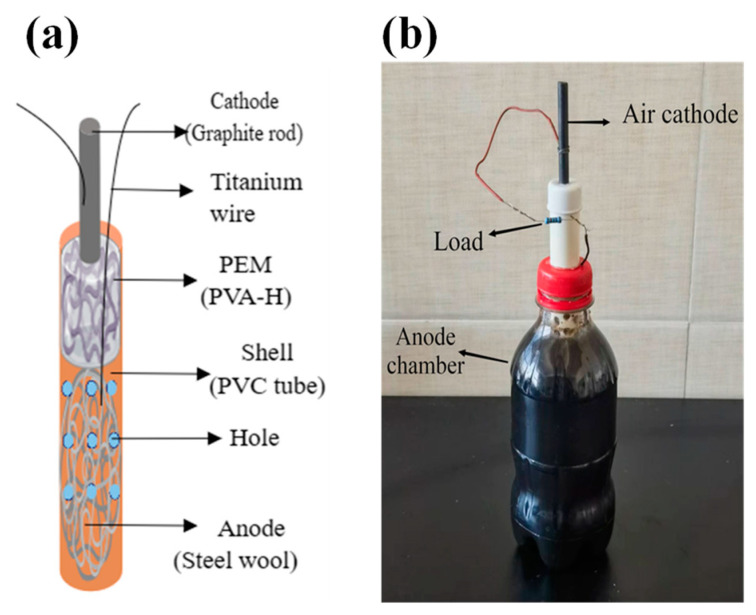
Schematic diagram (**a**) and physical picture (**b**) of tubular air-cathode microbial fuel cell.

**Figure 3 materials-17-01286-f003:**
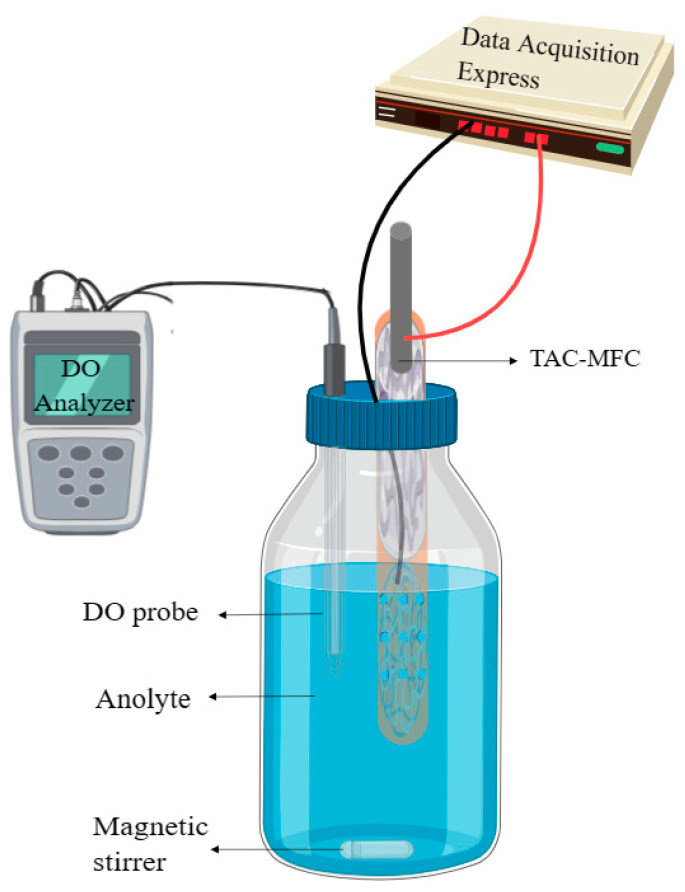
The diagram illustrating the oxygen permeation device.

**Figure 4 materials-17-01286-f004:**
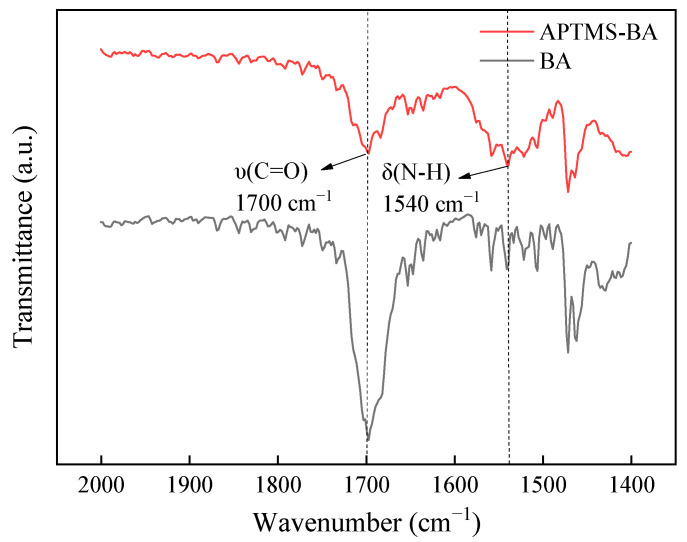
Fourier infrared spectra of BA and APTMS-BA were obtained with a wavenumber scanning range of 1400–2000 cm^−1^.

**Figure 5 materials-17-01286-f005:**
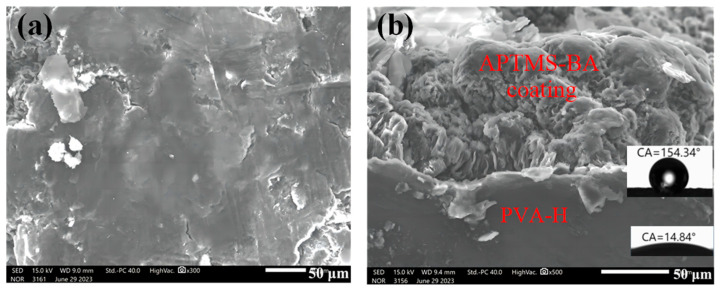
The scanning electron microscope images depict the AB-PVA-H, showcasing both (**a**) the surface and (**b**) the cross-section. The corresponding water contact angles (WCAs) of the surface are provided as inserts.

**Figure 6 materials-17-01286-f006:**
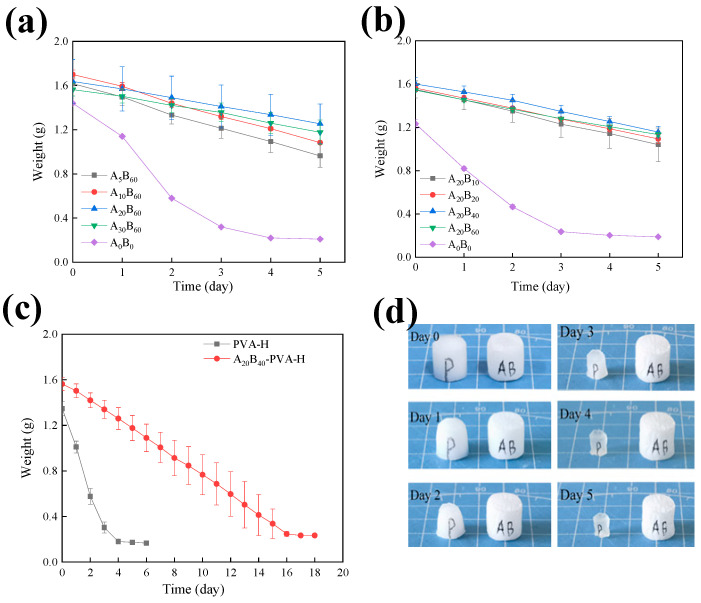
The change of hydrogel weight with time within 5 days in APTMS/H_2_O (**a**) and BA/n-Hexane (**b**) solutions with different modification times. (**c**) The weight of A_20_B_40_ coated hydrogel and PVA-H changed over 20 days with time. (**d**) The alteration in the appearance of both PVA-H (**left**) and AB-PVA-H (**right**) was observed over a period of 5 days (under experimental conditions of T = 25 °C and RH = 30%).

**Figure 7 materials-17-01286-f007:**
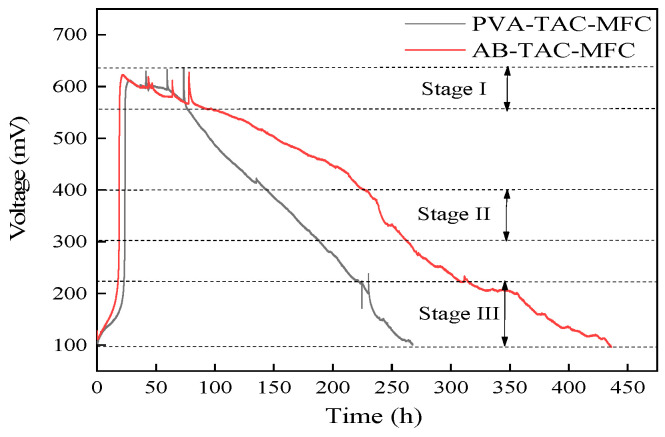
Voltage output of PVA-TAC-MFC (black line) and AB-TAC-MFC (red line).

**Figure 8 materials-17-01286-f008:**
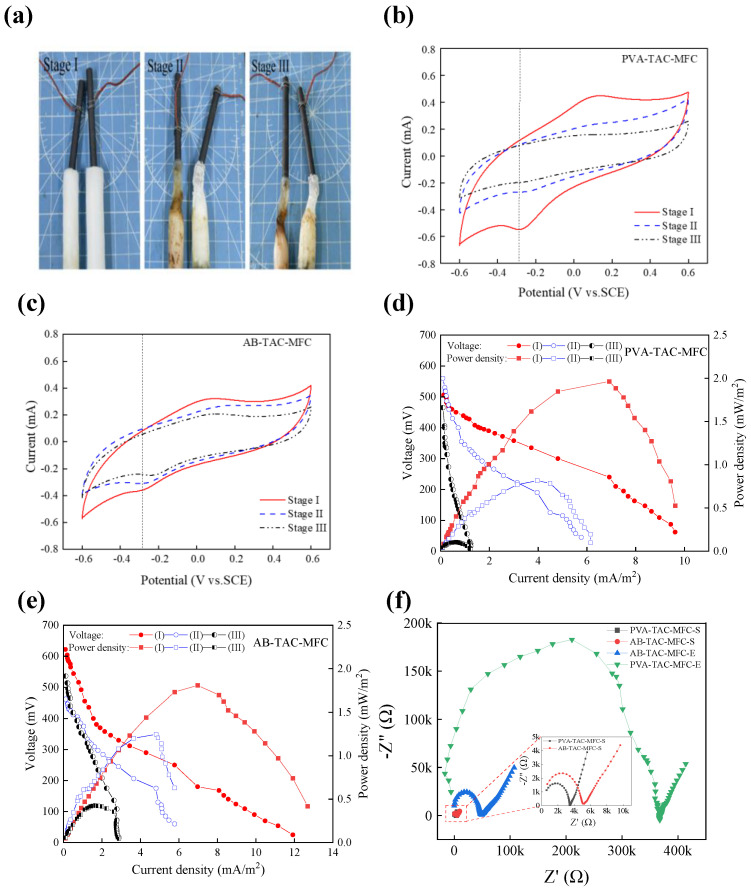
(**a**) The degree of contamination on both the PEM and cathode during three stages: PVA-H (left) and AB-PVA-H (right). CV curves of three stages for MFCs: (**b**) PVA-TAC-MFC, (**c**) AB-TAC-MFC (the vertical dashed line is the reduction peak). Plots of polarization curves and power densities curves across three stages: (**d**) PVA-TAC-MFC, (**e**) AB-TAC-MFC. (**f**) Changes in impedance were measured at the start (S) and end (E) of PVA-TAC-MFC and AB-TAC-MFC.

**Figure 9 materials-17-01286-f009:**
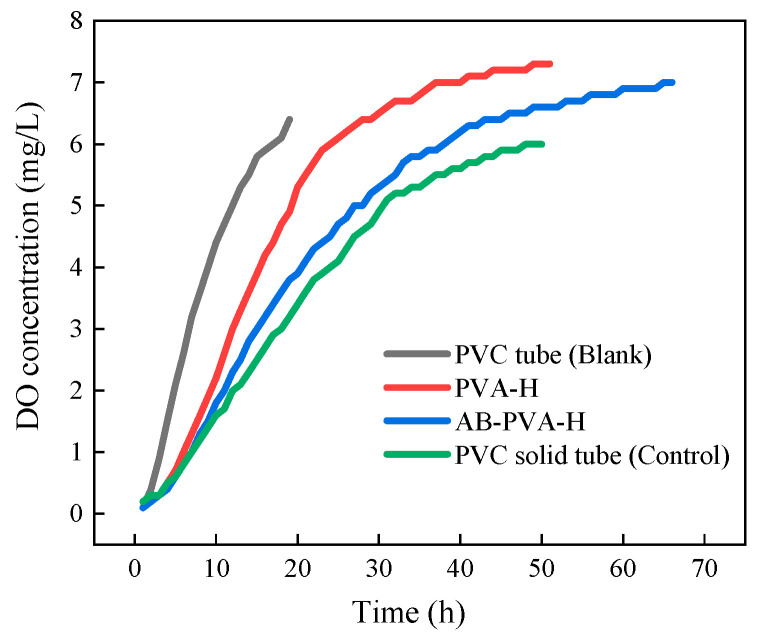
Variation in dissolved oxygen concentration in the anode solution across different types of proton exchange membranes.

**Table 1 materials-17-01286-t001:** Oxygen diffusion coefficients in the presence and absence of the proton exchange membrane.

Type of Proton Exchange Membrane	K (cm s^−1^)
PVC tube (Blank)	3.145 ± 0.003
PVA-H	2.402 ± 0.005
AB-PVA-H ^1^	1.859 ± 0.002
PVC solid tube (Control)	1.564 ± 0.003

^1^ AB-PVA-H: denoted as PVA-A_20_B_40_, signifies that A_20_ represents the modification time of the PVA-H in APTMS/H_2_O solution for 20 min, while B_40_ represents the modification time in BA/n-Hexane solution for 40 min.

**Table 2 materials-17-01286-t002:** The variation in degradation efficiency of chemical oxygen demand.

MFC Type	COD I (mg L^−1^)	COD II (mg L^−1^)	COD III (mg L^−1^)	ηCOD (%)
PVA-TAC-MFC	5370	5008	4496	16
AB-TAC-MFC	5472	4993	3566	35

## Data Availability

The data used in this research are properly cited and reported in the main text.

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
