# Peer review of "Enhancing the Proton Exchange Membrane in Tubular Air-Cathode Microbial Fuel Cells through a Hydrophobic Polymer Coating on a Hydrogel"

_materials, 2024, doi:10.3390/ma17061286_

Round 1

Reviewer 1 Report

Comments and Suggestions for Authors

This article introduces a method for fabricating a proton exchange membrane (PEM) based on polyvinyl alcohol hydrogel (PVA-H), which was incorporated into a tubular air cathode microbial fuel cell (AC-MFC). The surface of this PVA-H-PEM was modified with a hydrophobic polymer, leading to a longer water retention time compared to the non-modified PVA-H. This modification resulted in improved performance and operation time for the AC-MFC.

However, there are still some issues that need to be addressed.

Major:

The set up is not very clear to me. I am not sure about a solid graphite rod being used as an air cathode. More details on the air cathode are required; the reader cannot understand the basis of its function.

Power and current outputs are very low. Do the authors expect such low values? The internal resistances seem excessively high.

The EIS seems to show increasing size of the semi circle (which I do not think is necessarily charge transfer resistance), but the paper says RCT decreases. I would use power density curves to extract the internal resistance (using current and voltage at the maximum power point)

What do authors mean that the MFC “has an external resistance of 1000 ohms”? Do they mean that they set the external resistance to 1000 ohms? Please clarify. In any case, internal and external resistances should be matched, but the latter seems orders of magnitude smaller than the former.

Minor:

1)      The figures should be mentioned in sequential order. For instance, in the manuscript, there is a reference to Figure 1b followed by Figure 1a, and Figure 8 appears before Figure 7.

2)      Line 103 -113, these two paragraphs should be rewritten to be clearer. And some information is missing. For instance, although the preparation of the PVA hydrogel is mentioned in a PVC tube, the step at which the PVA hydrogel is removed from the tube is not discussed.

3)      The naming conventions throughout the paper need to be both distinct and consistent. Initially, 'PVA-H' is used to refer to the PVA hydrogel both before and after modification, which may lead to confusion for the reader. Towards the end, 'AB-PVA-H' is introduced to signify the modified form, which is a good approach. However, it's advisable to employ the same naming convention consistently throughout the entire paper for clarity and coherence. Inconsistency arises in the naming of the tubular air cathode microbial fuel cell (MFC) throughout the paper. At times, it is referred to as 'MFC,' while in other instances, 'AC-MFC' is used.

4)      Line 139-140, the statement implies that the anode chamber is not considered a part of the microbial fuel cell (MFC). However, it is important to note that the anode chamber is indeed a critical component of the MFC system.

5)      Line 157, it is stated that 'Stage I' represents the time when the voltage of AC-MFC rises from 100 mV to 600 mV. However, in Figure 7, the indicated voltage range for this stage is from 550 mV to 650 mV. It is essential to reconcile this inconsistency for accurate reporting and coherence between the text and figures.

6)      Line 162, it is noted that COD measurements were conducted for both Stage I and Stage III. However, it is not clear why measurements for Stage II were omitted.

7)      Line 187-188, the sentence, 'The polarization curve and power density curve were obtained by varying the external resistance.', appears to be in a position that might not align well with the surrounding context. Consider relocating this sentence to a more suitable position within the text to enhance coherence and clarity.

8)      Line 223, it is said that the peaks at 1540 and 1700 cm-1 are attributed to the amide I and amide II bands. However, it is crucial to clarify that these peaks likely belong to the amide band, as amide I and amide II are typically associated with protein structures.

9)      In Figure 4, if the aim is to demonstrate the successful grafting of BA to APTMS, it is advisable to include spectra for BA, APTMS, and the product BA-APTMS. Additionally, if the intention is to overlay these spectra for clarity and to prevent overlap, the appropriate unit for the y-axis should be 'a.u.' (arbitrary units) rather than '%'.

10)  In Figure 5b, it would be beneficial to label the BS side and APTMS side directly on the image for clarity.

11)  In Figure 6a-c, at time 0, according to equation 1, WRi should be 1 since wi = w0. However, the first data point in the graphs is approximately 1.6. It would be beneficial for the authors to provide clarification or explanation regarding this discrepancy to ensure the consistency and accuracy of the presented data.

12)  Line 355, there is mention of the amide bond (-CO-NH-), but it appears that the context is related to the polymer coating rather than the amide bond itself.

13)  In Figure 9, considering that the measurements were conducted over a period exceeding 2 days, it is important to take into account the volume changes of PVA.

14)  Chemical formulas should include subscripts: e.g., as in H2O

Overall, I suggest major revisions are required.

Comments on the Quality of English Language

Fine.

Author Response

Reviewer 1(reply to reviewers’ comments in blue color)

Major:

  1. The set up is not very clear to me. I am not sure about a solid graphite rod being used as an air cathode. More details on the air cathode are required; the reader cannot understand the basis of its function.

Reply: We sincerely thank you for careful reading and comments. Microbial fuel cells can be divided into two chambers and one chamber according to the structure, and can be divided into air cathode and non-air cathode according to the different electron acceptors of the cathode. Because the air cathode omits the cathode chamber, the oxygen in the air is directly used as the final electron acceptor, no waste is generated, and the cost is reduced. The traditional air cathode is mainly made of carbon cloth, the main component of which is graphite. Therefore, it is reasonable to use graphite rod as air cathode in the cylindrical tube air cathode microbial fuel cell designed in this study.

  1. Power and current outputs are very low. Do the authors expect such low values? The internal resistances seem excessively high.

Reply: Thanks for your comments.

  • Microbial fuel cells mainly rely on bacteria in the anode chamber to produce electricity, so its electricity generation performance is generally low. It is necessary to improve the electric generation performance of MFC by series or parallel connection of MFCs.
  • This experiment is to explore the difference between the voltage and power before and after the modification of the protonmembrane, or to explore the time of maximum power density maintenance and the time of voltage maintenance. Therefore, lower output power and current will not affect the content of the experiment.
  • It can be seen from the EIS that the internal resistance of the battery is indeed high. This is because PVA hydrogel is used in the proton exchange membrane of MFC in this experimental device. PVA hydrogel is a non-conductor, which can effectively avoid direct contact between cathode and anode. In addition, the water in the section where PVA hydrogel is in contact with the air cathode during the operation of the battery evaporates relatively quickly, resulting in increased resistance to proton transmission after the proton membrane dries. As a result, the impedance of MFC increases later in operation.
  1. The EIS seems to show increasing size of the semi circle (which I do not think is necessarily charge transfer resistance), but the paper says RCT decreases. I would use power density curves to extract the internal resistance (using current and voltage at the maximum power point)

Reply: We think this is an excellent suggestion.

  • According to your suggestion, we can see from Figure 8d and Figure 8e that the slope of polarization curves of MFC with polymer coated hydrogel as proton exchange membrane (AB-TAC-MFC) is lower than that of the battery coated with uncoated proton membrane (PVA-TAC-MFC) at the three stages of operation. The lower the slope of the polarization curve, the smaller the internal resistance of the MFC.
  • The arc radius of the Nyquist diagram corresponds to the charge transfer resistance (Rct), which can measure the internal resistance of the battery to a certain extent. Combined with EIS Figure 8f, the arc radius of the figure corresponds to the charge transfer impedance Rct, which is a part of the internal resistance of MFC. At the end stage of battery operation, the radius of the arc of the EIS curve (blue line) of AB-TAC-MFC is about 1/5 of that of the EIS curve (green line) of PVA-TAC-MFC.
  • From the above two points, it can be seen that the existence of polymer coating reduces the internal resistance of MFC.
  1. What do authors mean that the MFC “has an external resistance of 1000 ohms”? Do they mean that they set the external resistance to 1000 ohms? Please clarify. In any case, internal and external resistances should be matched, but the latter seems orders of magnitude smaller than the former.

Reply: Thanks for your comments. We're sorry it's not explained in the manuscript.

  • On the new line 163-164: The external 1000 ohms resistor is just a test resistor, which mainly tests the voltage output of the battery through the data collector under the load of 1000 ohms.
  • On the new line 172-176: When measuring the output power density, the 1000 ohm resistance will be replaced with a variable resistance box, and the maximum power density can be determined by adjusting the resistance value of the resistance box (99999~100 ohm), at this time the internal resistance of the battery and the number on the resistance box are equal.
  • We have added this passage to the manuscript and highlighted it with a red background.

Minor:

  1. The figures should be mentioned in sequential order. For instance, in the manuscript, there is a reference to Figure 1b followed by Figure 1a, and Figure 8 appears before Figure 7.

Reply: Thanks for your suggestions. We agree with your point and some revisions were made as follows:

  • On the new line 107-129:After the rewrite, the reference to Figure 1b followed by Figure 1a.
  • On the new line 301: This is a marked error, and “Figure 8” has been changed to “Figure 7” and highlighted it with a red background.
  1. Line 103 -113, these two paragraphs should be rewritten to be clearer. And some information is missing. For instance, although the preparation of the PVA hydrogel is mentioned in a PVC tube, the step at which the PVA hydrogel is removed from the tube is not discussed.

Reply: Thanks for your suggestions.

  • On the new line 107-129: We have rewritten this part according to your suggestion and highlighted it with a red background.
  • On the new line 111: Added information about the step at which the PVA hydrogel is removed from the tube.
  1. The naming conventions throughout the paper need to be both distinct and consistent. Initially, 'PVA-H' is used to refer to the PVA hydrogel both before and after modification, which may lead to confusion for the reader. Towards the end, 'AB-PVA-H' is introduced to signify the modified form, which is a good approach. However, it's advisable to employ the same naming convention consistently throughout the entire paper for clarity and coherence. Inconsistency arises in the naming of the tubular air cathode microbial fuel cell (MFC) throughout the paper. At times, it is referred to as 'MFC,' while in other instances, 'AC-MFC' is used.

Reply: Thanks for your suggestions.

  • According to your suggestion. “AB-coated-H” and “APTMS-BA-coated-H” have been replaced all with “AB-PVA-H” throughout the article and highlighted it with a red background.
  • On the new line 269: replaced “AxBy-PVA” with “AxBy-PVA-H” and highlighted it with a red background.
  • On the new line 272,279: replaced “A20B40-PVA” with “A20B40-PVA-H” and highlighted it with a red background.
  • On the new line 39:deleted “(AC-MFCs)” after “air-cathode microbial fuel cells”.
  1. Line 139-140, the statement implies that the anode chamber is not considered a part of the microbial fuel cell (MFC). However, it is important to note that the anode chamber is indeed a critical component of the MFC system.

Reply: Thanks for your suggestions.

  • On new line 148-149: revised the sentence of “The structure of the tubular air-cathode microbial fuel cell (TAC-MFC) is shown in Figure 2a” as “The structure of cathode anode and proton exchange membrane is shown in Figure 2a” and highlighted it with a red background.
  • On new line 155: It is explained here anode chamber is indeed a critical component of the MFC system and highlighted it with a red background.
  1. Line 157, it is stated that 'Stage I' represents the time when the voltage of AC-MFC rises from 100 mV to 600 mV. However, in Figure 7, the indicated voltage range for this stage is from 550 mV to 650 mV. It is essential to reconcile this inconsistency for accurate reporting and coherence between the text and figures.

Reply: Thanks for your suggestions. On new line 167: replaced “100 mV to 600 mV” with “550 mV to 650 mV” and highlighted it with a red background.

  1. Line 162, it is noted that COD measurements were conducted for both Stage I and Stage III. However, it is not clear why measurements for Stage II were omitted.

Reply: Thanks for your suggestions. On the new line 177: add “COD” before “polarization” and highlighted it with a red background. The measured value of COD at Stage II (COD II) can be seen from Table 2.

  1. Line 187-188, the sentence, 'The polarization curve and power density curve were obtained by varying the external resistance.', appears to be in a position that might not align well with the surrounding context. Consider relocating this sentence to a more suitable position within the text to enhance coherence and clarity.

Reply: Thanks for your suggestions.

  • Deleted the sentence of “The polarization curve and power density curve were obtained by varying the external resistance” on original line 187-188.
  • On the new line 172-173: add the sentence “The polarization curve and power density curve were obtained by varying the external resistance” after the sentence “At this point, the polarization curve, power density, and COD value of the TAC-MFC were measured”
  1. Line 223, it is said that the peaks at 1540 and 1700 cm-1are attributed to the amide I and amide II bands. However, it is crucial to clarify that these peaks likely belong to the amide band, as amide I and amide II are typically associated with protein structures.

Reply: Thanks for your suggestions. The amide I band and the amide II band are specific wave-number ranges in the infrared spectrum, which respectively represent two different modes of vibration in compounds containing the amide group (-CONH-) υ(C=O): the amide I band. It appears from 1700 to 1630cm-1, It shows the stretching vibration of the C=O group. δ(N-H):The amide II band is usually located in the range 1540-1655 cm-1, which reflects the stretching vibration of the NH group.

  1. In Figure 4, if the aim is to demonstrate the successful grafting of BA to APTMS, it is advisable to include spectra for BA, APTMS, and the product BA-APTMS. Additionally, if the intention is to overlay these spectra for clarity and to prevent overlap, the appropriate unit for the y-axis should be 'a.u.' (arbitrary units) rather than '%'.

Reply: Thanks for your suggestions.

  • We are sorry that we do not have the infrared spectral data of APTMS at present, which can be made up later in the paper.
  • In new figure 4:According to your suggestion, the Y-axis unit has been changed from "%" to "a.u.".
  1. In Figure 5b, it would be beneficial to label the BS side and APTMS side directly on the image for clarity.

Reply: Thanks for your suggestions. In new figure 5: According to your suggestion, we have marked the position of APTMS-BA coating and PVA-H in Figure 5b.

  1. In Figure 6a-c, at time 0, according to equation 1, WRi should be 1 since wi = w0. However, the first data point in the graphs is approximately 1.6. It would be beneficial for the authors to provide clarification or explanation regarding this discrepancy to ensure the consistency and accuracy of the presented data.

Reply: Thanks for your suggestions. We really sorry for our careless mistakes. In new Figure 6a-c: The ordinate is replaced by “Weight retention (g/g)” with “Weight (g)”, calculate WRi according to the data in the figure.

  1. Line 355, there is mention of the amide bond (-CO-NH-), but it appears that the context is related to the polymer coating rather than the amide bond itself.

Reply: Thanks for your suggestions. We agree with your points.

On the new line 370: deleted “(-CO-NH-)” after “polymer coating”

On the new line 399: deleted “represented by —CO—NH—" after “polymer coating”

  1. In Figure 9, considering that the measurements were conducted over a period exceeding 2 days, it is important to take into account the volume changes of PVA.

Reply: Thanks for your suggestions. We think this is an excellent suggestion. In order to maintain the anaerobic environment of the anode chamber, in this experimental device, the coating obstructs the diffusion of oxygen from the air cathode to the anode chamber in two aspects:

  • Due to the dense nature of the coating, it can prevent oxygen from entering the anode chamber through the PVA hydrogel.
  • Due to the volume of PVA gel will decrease after the water is lost, a large gap will be generated between PVA hydrogel and the round tube, which will easily lead to the direct entry of oxygen into the anode chamber. The coated hydrogel prevents the evaporation of internal water, and the volume of hydrogel will not be small so fast, so that there will not be a large gap between PVA hydrogel and the round tube. From this point of view, it also played a role in hindering the diffusion of oxygen to a certain extent.
  1. Chemical formulas should include subscripts: e.g., as in H2O

Reply: Thanks for your suggestions. We agree with your points.

On the new line 422: replaced “H2O” with “H2O” and highlighted it with a red background.

Reviewer 2 Report

Comments and Suggestions for Authors

This manuscript “Enhancing the proton exchange membrane in tubular air cathode microbial fuel cells through hydrophobic polymer coating on hydrogel” describes in detail the treatment of a microbial fuel cell in various steps to form a hydrophobic polymer coating layer onto the membrane material to enhance its durability by increasing the water retention rate. Performance data of a treated and an untreated membrane are compared, and it is found that the addition of the hydrophobic layer onto the otherwise hydrophilic membrane material leads to enhanced fuel cell lifetime at the cost of lower power density in the initial stage.

Overall, the manuscript is well written, despite an excessive use of terminology which for the layman will be difficult to understand. All treatment steps are described in detail so that this data should be fairy easily reproduced. The lifetime is increased by 2/3 (better write 66%), and the oxygen cross-over is reduced. The figures are generally of very good quality.

This manuscript can be published after several minor corrections have been made.

A central point seems to be that the performance of the microbial fuel cell is clearly limited by mass transport (Fig. 8e). Would it not be possible to provide a stirring to the anode solution, or otherwise enhance the oxygen transport by a fan?

Among the numerous lines that should be corrected are:

·        L. 35: hydrogen ions => protons

·        L. 68:  bracket

·        L. 77:  [28] added. Better use authors names

·        L. 142: (2 g s stainless wool),

·        L. 151

·        L. 154: 1000 Ω.The entire

·        L. 172: was used. dissolved oxygen

·        L. 182:  (1)[28], to assess

·        L. 212: the experiment, Given the provided

·        L. 256: and By represents the modification

·        L. 262-263: retention than A20B40 group 262 , likely due

·        L. 330: akin to findings by Li et al. [9].The power

·        L. 347: In Figure 8f, it's observed that in the

·        L. 390: coating alleviats the degree of oxygen

Comments on the Quality of English Language

This manuscript “Enhancing the proton exchange membrane in tubular air cathode microbial fuel cells through hydrophobic polymer coating on hydrogel” describes in detail the treatment of a microbial fuel cell in various steps to form a hydrophobic polymer coating layer onto the membrane material to enhance its durability by increasing the water retention rate. Performance data of a treated and an untreated membrane are compared, and it is found that the addition of the hydrophobic layer onto the otherwise hydrophilic membrane material leads to enhanced fuel cell lifetime at the cost of lower power density in the initial stage.

Overall, the manuscript is well written, despite an excessive use of terminology which for the layman will be difficult to understand. All treatment steps are described in detail so that this data should be fairy easily reproduced. The lifetime is increased by 2/3 (better write 66%), and the oxygen cross-over is reduced. The figures are generally of very good quality.

This manuscript can be published after several minor corrections have been made.

A central point seems to be that the performance of the microbial fuel cell is clearly limited by mass transport (Fig. 8e). Would it not be possible to provide a stirring to the anode solution, or otherwise enhance the oxygen transport by a fan?

Among the numerous lines that should be corrected are:

·        L. 35: hydrogen ions => protons

·        L. 68:  bracket

·        L. 77:  [28] added. Better use authors names

·        L. 142: (2 g s stainless wool),

·        L. 151

·        L. 154: 1000 Ω.The entire

·        L. 172: was used. dissolved oxygen

·        L. 182:  (1)[28], to assess

·        L. 212: the experiment, Given the provided

·        L. 256: and By represents the modification

·        L. 262-263: retention than A20B40 group 262 , likely due

·        L. 330: akin to findings by Li et al. [9].The power

·        L. 347: In Figure 8f, it's observed that in the

·        L. 390: coating alleviats the degree of oxygen

Author Response

Reviewer 2(reply to reviewers’ comments in blue color)

Comments and Suggestions for Authors

This manuscript “Enhancing the proton exchange membrane in tubular air cathode microbial fuel cells through hydrophobic polymer coating on hydrogel” describes in detail the treatment of a microbial fuel cell in various steps to form a hydrophobic polymer coating layer onto the membrane material to enhance its durability by increasing the water retention rate. Performance data of a treated and an untreated membrane are compared, and it is found that the addition of the hydrophobic layer onto the otherwise hydrophilic membrane material leads to enhanced fuel cell lifetime at the cost of lower power density in the initial stage.

Overall, the manuscript is well written, despite an excessive use of terminology which for the layman will be difficult to understand. All treatment steps are described in detail so that this data should be fairy easily reproduced. The lifetime is increased by 2/3 (better write 66%), and the oxygen cross-over is reduced. The figures are generally of very good quality.

Reply: Thanks for your comments.

On the new line 18,324,398: replaced “2/3” with “66%” and highlighted it with a blue background.

This manuscript can be published after several minor corrections have been made.

A central point seems to be that the performance of the microbial fuel cell is clearly limited by mass transport (Fig. 8e). Would it not be possible to provide a stirring to the anode solution, or otherwise enhance the oxygen transport by a fan?

Reply: We sincerely thank you for careful reading and comments. We think this is an excellent suggestion.

  • Due to protons are transported in the proton exchange membrane with water molecules as the carrier, the performance of microbial fuel cells is largely affected by the content of water in the proton exchange membrane. When the water in the membrane is dispersed, the proton transmission resistance becomes larger, the efficiency becomes lower, and the current transmission efficiency inside MFC decreases, thus reducing the battery's electricity generation performance.
  • The purpose of stirring the anodic solution is to make it easier for the electrogenic bacteria to attach to the anode, which can increase the contact area between them, thereby improving the electrogenic performance.
  • In order to maintain the anaerobic environment of the anode chamber, in this experimental device, The coating acts to reduce oxygen transport from the air cathode to the anode chamber.

Among the numerous lines that should be corrected are:

  1. 35: hydrogen ions => protons

Reply: Thanks for your comments.

On the new line 35: replaced “hydrogen ions” with “protons” and highlighted it with a blue background.

  1. 68: bracket

Reply: Thanks for your comments.

On the new line 72: deleted “(” before “[28]”

  1. 77: [28] added. Better use authors names

Reply: Thanks for your comments.

On the new line 81: added “Gao et al” before ”[30]” and highlighted it with a blue background.

  1. 142: (2 g s stainless wool),

Reply: Thanks for your comments.

On the new line 151: replaced “(2 g s stainless wool)” with ” (2 g stainless wool)” and highlighted it with a blue background.

  1. 151

Reply: Thanks for your comments.

On the new line 160:deleted “ ” before “ cro-”

  1. 154: 1000 Ω. The entire

Reply: Thanks for your comments.

On the new line 164: replaced “1000 Ω. The entire” with “ 1000 ohms. The entire” and highlighted it with a blue background.

  1. 172: was used. dissolved oxygen

Reply: Thanks for your comments.

On the new line 186 :replaced “dissolved” with “ Dissolved” and highlighted it with a blue background.

  1. 182: (1)[28], to assess

Reply: Thanks for your comments.

On the new line 196 : deleted “ ” before “ (1)[30]”

  1. 212: the experiment, Given the provided

Reply: Thanks for your comments.

On the new line 226: replaced “Given” with “given” and highlighted it with a blue background.

  1. 256: and By represents the modification

Reply: Thanks for your comments.

On the new line 270: replaced “By” with “by” and highlighted it with a blue background.

  1. 262-263: retention than A20B40 group 262, likely due

Reply: Thanks for your comments.

On the new line 276: deleted “ ” before “,”

  1. 330: akin to findings by Li et al. [9].The power

Reply: Thanks for your comments.

On the new line 345: added “ ” before “The”.

  1. 347: In Figure 8f, it's observed that in the

Reply: Thanks for your comments.

On the new line 362: replaced “In Figure 8f” with “According to Figure 8f” and highlighted it with a blue background.

  1. 390: coating alleviats the degree of oxygen

Reply: Thanks for your comments.

On the new line 405: replaced “alleviats” with “reduces” and highlighted it with a blue background.

Reviewer 3 Report

Comments and Suggestions for Authors

Referee´s comments

To the paper entitled

Enhancing the proton exchange membrane in tubular air cathode microbial fuel cells through hydrophobic polymer coating on hydrogel

By Junlin Huang et al.

The paper is devoted to very interesting problem, which is important from the practical point of view. In this work, materials for membrane fuel cell are reported. Microbial fuel cells combine  wastewater treatment and energy recovery. Over microbial degradation of organic compounds, chemical energy is converted into electrical energy.

In order to decrease the efficiency of proton exchange membrane, it is proposed to coat it with hydrophobic polymer. This coating prevents a leakage of anolyte into the air cathode.

The paper should be published earlier or later after some important additions and corrections.

First of all, the scheme, which illustrates the operation principle of microbial fuel cell, is very welcome.

``The efficiency of proton transport within PEM strongly relies on its water content. `` Please mention that one possible way ty increase water content in proton conducting polymers is their modifying with inorganic ion exchangers (https://doi.org/10.1134/S1023193513030075,). But their effect is ambiguous, the modifier can both increase and reduce water uptake (https://doi.org/10.1016/j.memsci.2018.06.024, https://doi.org/10.1134/S1023193513030075). This behavior is caused by morphology of embedded inorganic particles. That is the advantage of your approach that avoids embedding inorganic particles inside ion exchange polymer.

What anaerobic bacteria were collected from waste waters?

Fig. 3. How the concentration of dissolved oxygen was determined? In general, it is very low. What is an electronic device for determining oxygen content? Point also a producing company.   

``The COD of the anode liquid was determined through the fast digestion-spectrophotometric method.`` Please give the devices and producing companies.

Fig. 4. Please show the spectra within the interval of  4000-400 cm-1. Especially the region of 4000-3000 cm-1 is interesting, since a stripe due to water vibration should be absent.

  Fig. 8. Compare your results for power density with literature data for fuel cell containing polymer or polymer-inorganic proton conducting membrane.

Fig. 9. I would say that the oxygen concentration for AB-coated H is the same than that for the PVC, however it increases slower. Please note this in the text.

Comments on the Quality of English Language

English must be slightly improved.

Author Response

Reviewer 3(reply to reviewers’ comments in blue color)

Enhancing the proton exchange membrane in tubular air cathode microbial fuel cells through hydrophobic polymer coating on hydrogel

By Junlin Huang et al.

The paper is devoted to very interesting problem, which is important from the practical point of view. In this work, materials for membrane fuel cell are reported. Microbial fuel cells combine  wastewater treatment and energy recovery. Over microbial degradation of organic compounds, chemical energy is converted into electrical energy.

In order to decrease the efficiency of proton exchange membrane, it is proposed to coat it with hydrophobic polymer. This coating prevents a leakage of anolyte into the air cathode.

The paper should be published earlier or later after some important additions and corrections.

First of all, the scheme, which illustrates the operation principle of microbial fuel cell, is very welcome.

``The efficiency of proton transport within PEM strongly relies on its water content. `` Please mention that one possible way ty increase water content in proton conducting polymers is their modifying with inorganic ion exchangers (https://doi.org/10.1134/S1023193513030075,). But their effect is ambiguous, the modifier can both increase and reduce water uptake (https://doi.org/10.1016/j.memsci.2018.06.024, https://doi.org/10.1134/S1023193513030075). This behavior is caused by morphology of embedded inorganic particles. That is the advantage of your approach that avoids embedding inorganic particles inside ion exchange polymer.

Reply: We sincerely thank you for careful reading and comments.

On the new line 65-68 : Add references to literature and the sentence of “The polarization curve and power density curve were obtained by varying the external resistance” after “a key challenge” and highlighted it with a green background.

On the new line 69-70: Add the sentence of “It can avoids embedding inorganic particles inside ion exchange polymer.” After “prevent dehydration” and highlighted it with a green background.

What anaerobic bacteria were collected from waste waters?

Reply: Thanks for your comments. The microorganisms in the anodic solution were mainly anaerobic photosynthetic bacteria domesticated with kitchen waste.

Fig. 3. How the concentration of dissolved oxygen was determined? In general, it is very low. What is an electronic device for determining oxygen content? Point also a producing company.

Reply: Thanks for your comments. the concentration of dissolved oxygen in water is measured by the SMART AR8406 dissolved oxygen meter. It is produced by the Chinese company SMART.

On the new line 144-145: Added equipment and company information about dissolved oxygen meter equipment and highlighted it with a green background.

``The COD of the anode liquid was determined through the fast digestion-spectrophotometric method.`` Please give the devices and producing companies.

Reply: Thanks for your comments.

On the new line 201-202: Added equipment and company information for water analyzers and rapid digesters and highlighted it with a green background.

Fig. 4. Please show the spectra within the interval of  4000-400 cm-1. Especially the region of 4000-3000 cm-1 is interesting, since a stripe due to water vibration should be absent.

Reply: Thanks for your comments. We are sorry that we do not have the infrared spectral data of 4000-400 cm-1 at present, which can be made up later in the paper. The amide I band and the amide II band are specific wave-number mainly ranges(2000-1400 cm-1) in the infrared spectrum, which respectively represent two different modes of vibration in compounds containing the amide group (-CONH-). υ(C=O): the amide I band. It appears from 1700 to 1630cm-1, It shows the stretching vibration of the C=O group. δ(N-H):The amide II band is usually located in the range 1540-1655 cm-1, which reflects the stretching vibration of the NH group.

Fig. 8. Compare your results for power density with literature data for fuel cell containing polymer or polymer-inorganic proton conducting membrane.

Reply: Thanks for your suggestions. 

  • Microbial fuel cells mainly rely on bacteria in the anode chamber to produce electricity, so its electricity generation performance is generally low. It is necessary to improve the electric generation performance of MFC by series or parallel connection of MFCs.
  • This experiment is to explore the difference between the voltage and power before and after the modification of the protonmembrane, or to explore the time of maximum power density maintenance and the time of voltage maintenance. Therefore, lower output power and current will not affect the content of the experiment.

Fig. 9. I would say that the oxygen concentration for AB-coated H is the same than that for the PVC, however it increases slower. Please note this in the text.

Reply: Thanks for your suggestions. We think this is an excellent suggestion. In order to maintain the anaerobic environment of the anode chamber, in this experimental device, the coating obstructs the diffusion of oxygen from the air cathode to the anode chamber in two aspects:

  • Due to the dense nature of the coating, it can prevent oxygen from entering the anode chamber through the PVA hydrogel, therefor it increases slower.
  • Due to the volume of PVA hydrogel will decrease after the water is lost, a large gap will be generated between PVA hydrogel and the round tube, which will easily lead to the direct entry of oxygen into the anode chamber. The coated hydrogel prevents the evaporation of internal water, and the volume of hydrogel will not be small so fast, so that there will not be a large gap between PVA hydrogel and the round tube. From this point of view, it also played a role in hindering the diffusion of oxygen to a certain extent,therefor it increases slower.

Round 2

Reviewer 1 Report

Comments and Suggestions for Authors

The authors have responded to my review. Minor comments were addressed fairly. The major comments that cut to major problems with the system and analysis still persist and prevent me from recommending this work for publication. In my opinion, the authors must present their work in a way that enables proper evaluation of their work (e.g., proper normalized powers and measurements of the internal resistances). This will likely reveal outputs are far too low, because of internal resistances which are far too high. In this case, I do not find the work compelling because we cannot determine how effective the PEM is (the focus of this work) when it is superimposed on such a poorly performing device.

In advance, IK appologize for the use of all caps, which i found convenient to differentiate current review items from previous ones when I was preparing my review. In any case, I have also highlighted them.

ORIGINAL COMMENT 1: The set up is not very clear to me. I am not sure about a solid graphite rod being used as an air cathode. More details on the air cathode are required; the reader cannot understand the basis of its function.

AUTHOR’S REPLY TO COMMENT 1: We sincerely thank you for careful reading and comments. Microbial fuel cells can be divided into two chambers and one chamber according to the structure, and can be divided into air cathode and non-air cathode according to the different electron acceptors of the cathode. Because the air cathode omits the cathode chamber, the oxygen in the air is directly used as the final electron acceptor, no waste is generated, and the cost is reduced. The traditional air cathode is mainly made of carbon cloth, the main component of which is graphite. Therefore, it is reasonable to use graphite rod as air cathode in the cylindrical tube air cathode microbial fuel cell designed in this study.

RESPONSE TO AUTHOR’S REPLY 1:

The authors’ response and modifications to the paper are not acceptable. An air cathode must be porous to allow air to pass to the liquid/air interface (with the aid of a membrane) where the reaction can take place. If the solid electrode is immersed into the gel, then this might work, but as I stated initially, the reader still cannot understand the basis of operation. For example the authors write: “…followed by PEM (a PVA-H with a diameter of 16 mm and a length of 8 cm) extending to the top of the tube and embedded into a graphite rod (inner diameter of 5 mm, length of 5 cm).”

Is the PEM really embedded into the graphite? If the rod is embedded in the PEM (more likely) how deep is it embedded and how much surface area is in contact with the PEM? These are critical details.

REJECT AND RESUBMIT AFTER A CLEARER EXPLANATION IS GIVEN AND/OR THE EXPERIMENT IS REDONE WITH A REAL AIR-CATHODE. ONE RECOMMENDATION BEFORE SWITICHING ELECTRODES IS TO INSERT THE ROD DEEPER INTO THE PEM, IN ORDER TO INCREASE THE SURFACE AREA OF THE FUNCTIONAL PART OF THE ELECTRODE.

ORIGINAL COMMENT 2: Power and current outputs are very low. Do the authors expect such low values? The internal resistances seem excessively high.

AUTHOR’S REPLY TO COMMENT 2: Thanks for your comments.

  • Microbial fuel cells mainly rely on bacteria in the anode chamber to produce electricity, so its electricity generation performance is generally low. It is necessary to improve the electric generation performance of MFC by series or parallel connection of MFCs.

PARTIAL RESPONSE TO AUTHOR’S REPLY 2:

This is not a good response. My comment about low power is in the context of MFCs. This system appears to me to be highly unoptimized. Making it difficult to judge the PEM element performance.

VERDICT: REJECT AND RESUBUMIT AFTER PROPERLY PRESENTING OUTPUTS BY PROPER NORMALIZATION SHOW REASONABLE OUTPUTS FOR AN MFC.

  • This experiment is to explore the difference between the voltage and power before and after the modification of the protonmembrane, or to explore the time of maximum power density maintenance and the time of voltage maintenance. Therefore, lower output power and current will not affect the content of the experiment.

PARTIAL RESPONSE TO AUTHOR’S REPLY 2:

It is difficult to evaluate the performance of the PEM, if there are other major bottlenecks in the setup. The results should be normalised the anode and/or cathode surface area (important, this should be the portion that is in contact with the PEM). Normalizing by the volume provides no insights into the device performance.

VERDICT: PRESENT OUTPUTS WITH PROPER ELECTRODE AREA NORMALIZATION. COMPARE THIS TO THE LITERATURE. ASSUMING OUTPUTS ARE EXCESSIVELY LOW (PREVENTING A GOOD BASIS TO EVALUATE THE PEM, REDO EXPERIMENT AFTER OPTIMIZING AIR CATHODE)

  • It can be seen from the EIS that the internal resistance of the battery is indeed high. This is because PVA hydrogel is used in the proton exchange membrane of MFC in this experimental device. PVA hydrogel is a non-conductor, which can effectively avoid direct contact between cathode and anode. In addition, the water in the section where PVA hydrogel is in contact with the air cathode during the operation of the battery evaporates relatively quickly, resulting in increased resistance to proton transmission after the proton membrane dries. As a result, the impedance of MFC increases later in operation.

PARTIAL RESPONSE TO AUTHOR’S REPLY 2:

The response seems to indicate some knowledge gaps. No fuel cell (or MFC) has a membrane that allows direct contact between anode and cathode, as that would lead to an internal short-circuit. I assert that the nature of the air cathode is the problem.

ORIGINAL COMMENT 3: The EIS seems to show increasing size of the semi circle (which I do not think is necessarily charge transfer resistance), but the paper says RCT decreases. I would use power density curves to extract the internal resistance (using current and voltage at the maximum power point)

AUTHOR’S PARTIAL REPLY TO COMMENT 3: We think this is an excellent suggestion.

  • According to your suggestion, we can see from Figure 8d and Figure 8e that the slope of polarization curves of MFC with polymer coated hydrogel as proton exchange membrane (AB-TAC-MFC) is lower than that of the battery coated with uncoated proton membrane (PVA-TAC-MFC) at the three stages of operation. The lower the slope of the polarization curve, the smaller the internal resistance of the MFC.

VERDICT TO AUTHORS PARTIAL RESPONSE TO COMMENT 3: This is too qualitative. Please calculate the slope and give us the internal resistances. I think this will show an unacceptably high value, which I am sure is related to the construction of the cathode.

AUTHOR’S PARTIAL REPLY TO COMMENT 3:

·         The arc radius of the Nyquist diagram corresponds to the charge transfer resistance (Rct), which can measure the internal resistance of the battery to a certain extent. Combined with EIS Figure 8f, the arc radius of the figure corresponds to the charge transfer impedance Rct, which is a part of the internal resistance of MFC. At the end stage of battery operation, the radius of the arc of the EIS curve (blue line) of AB-TAC-MFC is about 1/5 of that of the EIS curve (green line) of PVA-TAC-MFC.

VERDICT TO AUTHORS PARTIAL RESPONSE TO COMMENT 3: THERE IS AN OVER FOCUS ON RCT, WHICH AS THE AUTHORS STATE IS ONLY A PARTIAL DRIVER OF OVERALL INTERNAL RESISTANCE. IN FACT, THE DIFFUSION RESISTANCE IS JUST AS IMPORTANT (IF NOT MORE FOR THIS APPLICATION). IN THE END THE TOTAL INTERNAL RESISTANCE IS MORE IMPORANT. SEE ABOVE.

  • From the above two points, it can be seen that the existence of polymer coating reduces the internal resistance of MFC.

VERDICT TO AUTHORS PARTIAL RESPONSE TO COMMENT 3: IT IS HARD TO MAKE ANY CONCLUSIONS ABOUT THE PEM WHEN THE OUTPUTS ARE SO LOW.

Comments on the Quality of English Language

Fine.

Reviewer 3 Report

Comments and Suggestions for Authors

As for my opinion, the revised paper looks more attractive than the initial version. THe paper could be published.

Comments on the Quality of English Language

English is more or less acceptable.

Author Response

Thanks

Round 3

Reviewer 1 Report

Comments and Suggestions for Authors

They authors have provided their response. I am not completely satisfied with their responses, but some of my comments have led to advancements. Under the condition that the authors calculate and present their outputs by normalizing by the contact area of the cathode with the PEM (1.77 cm2), and not by volume normalization, this can be published. Otherwise, I reject this paper.

Comments on the Quality of English Language

It's fine.

Author Response

Please review the attached file for details regarding the response to the question.
